# Direct in-situ insights into the asymmetric surface reconstruction of rutile TiO$_2$ (110)

Wentao Yuan[1,2,5], Bingwei Chen[1,5], Zhong-Kang Han ●[1,3] ✉, Ruiyang You[1], Ying Jiang[1], Rui Qi[1], Guanxing Li[1], Hanglong Wu ●[1], Maria Veronica Ganduglia-Pirovano ●[4] & Yong Wang ●[1] ✉

The reconstruction of rutile TiO$_2$ (110) holds significant importance as it profoundly influences the surface chemistry and catalytic properties of this widely used material in various applications, from photocatalysis to solar energy conversion. Here, we directly observe the asymmetric surface reconstruction of rutile TiO$_2$ (110)·(1×2) with atomic-resolution using in situ spherical aberration-corrected scanning transmission electron microscopy. Density functional theory calculations were employed to complement the experimental observations. Our findings highlight the pivotal role played by repulsive electrostatic interaction among the small polarons −formed by excess electrons following the removal of neutral oxygen atoms− and the subsequent surface relaxations induced by these polarons. The emergence and disappearance of these asymmetric structures can be controlled by adjusting the oxygen partial pressure. This research provides a deeper understanding, prediction, and manipulation of the surface reconstructions of rutile TiO$_2$ (110), holding implications for a diverse range of applications and technological advancements involving rutile-based materials.

Titanium dioxide (TiO$_2$) holds strategic significance in various applications, particularly as a photocatalyst for environmental remediation and the photoelectrochemical conversion of solar energy[1–4]. The functional properties of TiO$_2$ materials are intricately linked to their surface properties, driving extensive research efforts to understand the atomic and electronic structure of TiO$_2$ surfaces[5–21]. It is well-known that oxide surfaces, including rutile TiO$_2$ (110), undergo reconstructions under specific reaction conditions, such as high temperatures[22,23]. The rutile TiO$_2$ (110) surface has been reported to reconstruct into TiO$_2$ (110)·(1×1)[24–26], TiO$_2$ (110)·(1×2)[10,27,28], and pseudo-hexagonal rosette structures[29], depending on the oxygen chemical potential, $\mu_{O_2}(T,p)$. However, the precise atomic arrangement in these reconstructions and their formation mechanisms remain subjects of debate. This is in part due to a lack of consistency between both experimental and theoretical results. Specifically, for the TiO$_2$ (110)·(1×2) reconstruction,

various geometric structures have been proposed in experimental studies[10,22,24,25,27–31], yet these experiments lack the ability to provide an unambiguous description of atomic positions or surface stoichiometry. Single-linked and cross-linked TiO$_2$ (110)·(1×2) reconstructions have been observed following sputtering and annealing of samples in high vacuum at various temperatures[10,27,28,30]. The Ti$_2$O$_3$-(1×2) model, originally proposed by Onishi et al.[27,30], has been considered for the single-linked reconstruction, favored for its compatibility with low-energy electron diffraction results[28,31]. However, Park et al.[24,25] have proposed an alternative single-linked Ti$_2$O-(1×2) model, supported by transmission electron microscopy measurements[22]. Furthermore, a Ti$_3$O$_6$-(1×2) model, mirroring the bulk system's stoichiometry, has been used to describe the cross-linked TiO$_2$ (110)·(1×2) reconstruction[10]. Nevertheless, this model has been challenged, as the formation of the cross-linked TiO$_2$ (110)·(1×2) reconstruction requires

[1]Center of Electron Microscopy and State Key Laboratory of Silicon Materials, School of Materials Science and Engineering, Zhejiang University, 310027 Hangzhou, China. [2]Shanxi-Zheda Institute of Advanced Materials and Chemical Engineering, 030000 Taiyuan, China. [3]Fritz Haber Institute of the Max Planck Society, Faradayweg 4-6, 14195 Berlin, Germany. [4]Institute of Catalysis and Petrochemistry, ICP-CSIC, C/Marie Curie 2, 28049 Madrid, Spain. [5]These authors contributed equally: Wentao Yuan, Bingwei Chen. ✉e-mail: hanzk@zju.edu.cn; yongwang@zju.edu.cn

higher temperatures than the single-linked and thus is expected to be more oxygen deficient than the $Ti_2O_3$-(1×2) model[12]. To address these controversies, theoretical calculations combined with global optimization algorithms were employed to investigate the thermodynamically stable atomic configurations of $TiO_2$ (110)-(1×2) reconstructions[12]. One $Ti_2O_3$-(1×2) and one $Ti_3O_2$-(1×2) structure were identified to correspond to the reported single-linked and cross-linked $TiO_2$ (110)-(1×2) reconstructions, respectively[12]. In the meanwhile, the global optimization method[12] and studies using total reflection high-energy positron diffraction[32] suggest the existence of an asymmetric $Ti_2O_3$-(1×2) reconstruction driven by polarons[13]. However, direct observation of the symmetry of the $Ti_2O_3$-(1×2) reconstruction remains elusive, primarily due to the lack of side-view information. The relative stability of the symmetric and asymmetric $Ti_2O_3$-(1×2) reconstructions and their formation mechanisms have yet to be resolved. Scanning/transmission electron microscopy (S/TEM) emerges as a key tool for studying surface reconstructions from both top-view and side-view angles[6], providing sub-layer two-dimensional projection data that has previously helped in revealing the atomic structure and reconstruction mechanism of the anatase $TiO_2$ (001)-(1×4) reconstruction[33].

In this study, we provide a comprehensive understanding of the $TiO_2$ (110)-(1×2) surface reconstruction by employing in situ spherical aberration (Cs)-corrected scanning TEM (STEM) in combination with density functional theory calculations. This approach enables the precise determination of the positions of surface Ti and O atomic columns on the reconstructed $TiO_2$ (110) surface. Our work offers experimental evidence confirming the presence of the $Ti_2O_3$-(1×2) asymmetric reconstruction, lending support to previous suggestions[12,32]. The formation mechanism of this reconstruction is elucidated in terms of polaronic effects, highlighting the critical role played by the distributions of polarons. Additionally, we demonstrate the high sensitivity of the asymmetric reconstructions, allowing for modulation through fine-tuning of the oxygen partial pressure.

## Results and discussion
### HAADF STEM images of the rutile $TiO_2$ (110) surface
Upon heating the sample to 900 °C in a vacuum, the bulk-truncated rutile $TiO_2$ (110)-(1×1) surface undergoes reconstruction. High-angle annular dark field (HAADF) STEM profile images along the [001] zone axis, as shown in Fig. 1a, reveal numerous regularly spaced protrusive bright-dot pairs on the (110) surface. These bright dots correspond to the Ti atomic protrusions, supported by the $Z$-contrast of the HAADF STEM image, where $Z$ represents the atomic number[34,35]. The spacing between consecutive Ti column pairs measures approximately 1.31 nm, confirming the expected (1×2) reconstruction. In the bright-field (BF) STEM image (Fig. 1b), the Ti columns appear as dark-dot pairs. A closer examination of this BF STEM image, particularly in the magnified portion in Fig. 1c, reveals three protruding O columns (slightly brighter

dark dots, highlighted with red arrows) positioned between the Ti column pair and the subsurface layer. These observed features align well with the $Ti_2O_3$-(1×2) reconstruction model. However, an intriguing discovery is the varying heights of the Ti columns within each pair, with differences ranging from 0.02 to 0.07 nm. This suggests an inherent asymmetry in each pair of protruding Ti rows on the reconstructed $TiO_2$ (110)-(1×2) surface.

We conducted experiments to explore the intrinsic properties of the (1×2) reconstructed rutile $TiO_2$ (110) surfaces, focusing on the evolution of surface structures in various environments. Initially, the $TiO_2$ (110) surface exhibits a (1×1) structure when heated to 700 °C under an oxygen pressure of $2.53 \times 10^{-3}$ Pa, as shown in Supplementary Fig. 1a. Upon evacuating the $O_2$ gas, the (1×2) reconstruction progressively emerges on the surface (illustrated in Supplementary Fig. 1b, c). At the onset of the reconstruction, three $TiO_x$ double rows can be observed within the dotted rectangles, presenting a 3d-3d pattern (with "d" denoting the periodicity of the 1 × 1 bulk-terminated surface). As the reconstruction unfolds, the central double row gradually bifurcates into two distinct double rows, resulting in a 1d-1d-1d pattern. Supplementary Fig. 1d provides intensity profiles of the top layer, facilitating a more in-depth analysis of the surface structure.

### Symmetric and asymmetric $Ti_2O_3$-(1×2) reconstructions
To support the experimental observations, we performed systematic DFT calculations. Given the existence of multiple distributions of polarons in the near-surface region of reduced $TiO_2$[36], we explored different distributions of $Ti^{3+}$ polarons within the protruding reconstruction, the top two titanium layers, and selected deeper titanium layers (Supplementary Fig. 2). The four $Ti^{3+}$ ions of the $Ti_2O_3$-(1×2) reconstructions tend to stay near the surface and separate due to repulsive electrostatic interactions. Symmetric and asymmetric $Ti_2O_3$-(1×2) reconstructions were identified, with the asymmetric reconstruction proving more stable by approximately 0.3 eV, consistent with our in situ STEM observations, where most $Ti_2O_3$-(1×2) reconstructions are asymmetric. At high temperatures, vibrational entropy can influence the relative stability of different structural configurations of material surfaces[37]. The free energy differences of the symmetric and asymmetric $Ti_2O_3$-(1×2) reconstructions at different temperatures, incorporating vibrational entropy, are presented in Supplementary Table 1. The thermodynamic stability of the asymmetric reconstruction is even higher at elevated temperatures due to its disorder-mediated higher vibrational entropy. The most stable configurations of symmetric and asymmetric $Ti_2O_3$-(1×2) reconstructions are shown in Fig. 2. In the symmetric $Ti_2O_3$-(1×2) reconstruction (Fig. 2a), the two columns of the titanium protrusion have the same height relative to the substrate. In contrast, the asymmetric $Ti_2O_3$-(1×2) reconstruction (Fig. 2b) exhibits a lower titanium column relaxed

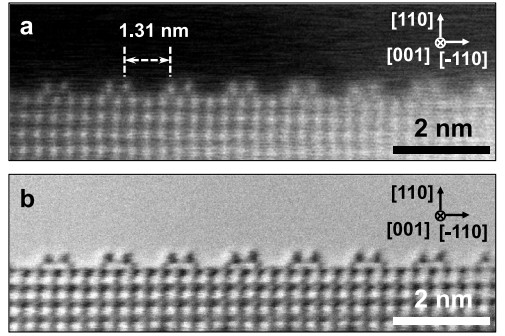
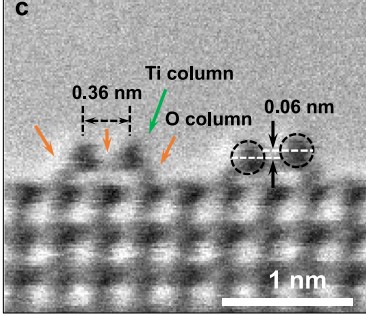
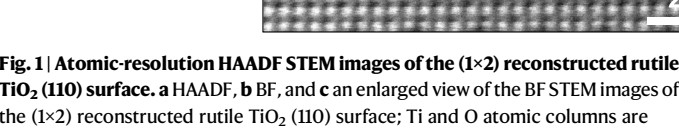

**Fig. 1 | Atomic-resolution HAADF STEM images of the (1×2) reconstructed rutile $TiO_2$ (110) surface. a** HAADF, **b** BF, and **c** an enlarged view of the BF STEM images of the (1×2) reconstructed rutile $TiO_2$ (110) surface; Ti and O atomic columns are indicated by the green and orange arrows, respectively. These images are acquired in situ at 900 °C in vacuum (≈ $10^{-5}$ Pa).

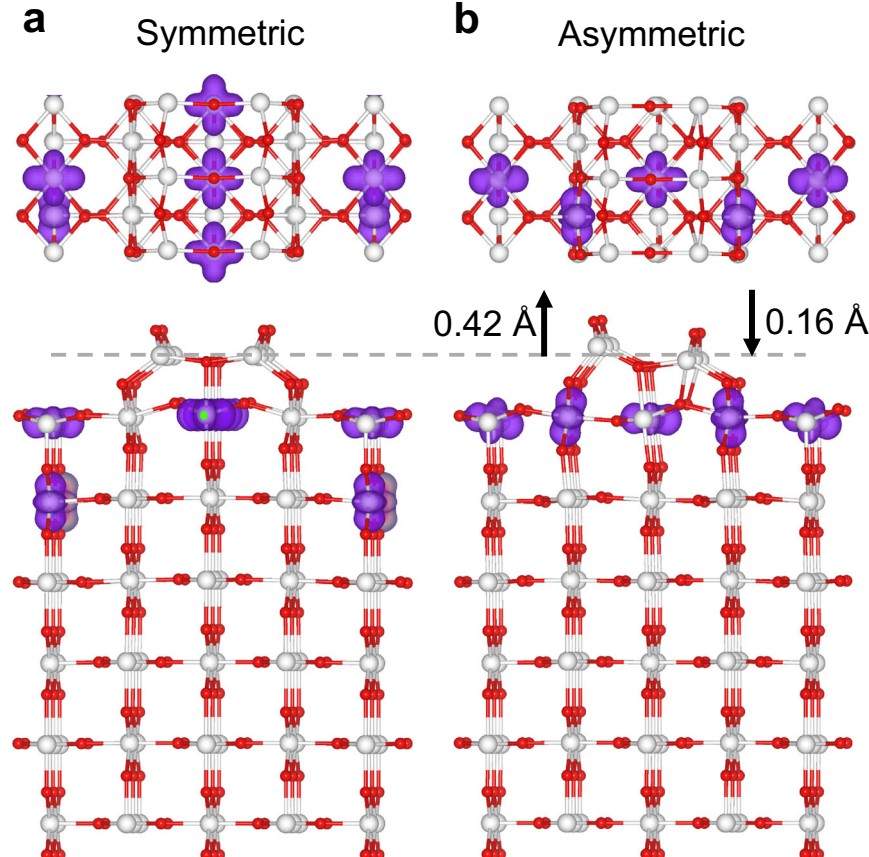

**a** Symmetric    **b** Asymmetric

0.42 Å    0.16 Å

**Fig. 2 | Structural model of the Ti$_2$O$_3$-(1×2) reconstructions.** Symmetric **a** and asymmetric **b** Ti$_2$O$_3$-(1×2) reconstructions together with the spin-charge density distributions. The titanium and oxygen atoms are represented by white and red balls, respectively.

downward by 0.16 Å and a higher titanium column relaxed upward by 0.42 Å relative to the symmetric Ti$_2$O$_3$-(1×2) reconstruction. These structural characteristics align well with our experimental STEM images, although the oxygen atoms in the outermost layer were not distinctly visible in our experiments due to resolution constraints. To assess whether the observed atomic structure aligns with the Ti$_2$O-(1×2) model proposed by Park et al.[24], we performed calculations on this model. Various configurations of Ti$^{3+}$ were considered for determining the most stable Ti$^{3+}$ distribution, as depicted in Supplementary Fig. 3. Phase diagrams plotting surface formation energies per unit cell for both the Ti$_2$O$_3$-(1×2) reconstruction model and the Ti$_2$O-(1×2) model under varying oxygen chemical potentials, relative to the unreconstructed surface, are presented in Supplementary Fig. 4. These diagrams reveal that with decreasing oxygen chemical potential (or at constant oxygen partial pressure with increasing temperature), the TiO$_2$ (110) surface undergoes a transition from the unreconstructed surface to the Ti$_2$O$_3$-(1×2) reconstruction model, consistent with earlier theoretical studies[12,24]. Under our experimental conditions, the Ti$_2$O$_3$-(1×2) reconstruction model proves more stable than the Ti$_2$O-(1×2) model. Additionally, our detailed calculations reveal that the Ti$_2$O-(1×2) model is inherently symmetric, with its asymmetric configuration being unstable and reverting to a symmetric form following first-principles structural optimization. Consequently, the asymmetric configuration observed in our experiments is not representative of the Ti$_2$O-(1×2) model, but rather of the Ti$_2$O$_3$-(1×2) reconstruction model.

We then explore the underlying mechanism for the formation of the asymmetric Ti$_2$O$_3$-(1×2) reconstruction. Building upon previous studies that highlighted the significant influence of orbital-dependent polaron-polaron interactions on system stability[38,39], our initial investigation focuses on the role of polaronic effects in the formation of Ti$_2$O$_3$-(1×2) reconstructions. Interestingly, our findings reveal that for the most stable asymmetric Ti$_2$O$_3$-(1×2) reconstruction (Fig. 2b), all four Ti$^{3+}$ ions are exclusively located in the first titanium layer. In contrast, in the case of the most stable symmetric Ti$_2$O$_3$-(1×2) reconstruction (Fig. 2a), three Ti$^{3+}$ ions reside in the first titanium layer, while one is located in the second titanium layer. The distinct distribution patterns of Ti$^{3+}$ ions in symmetric and asymmetric Ti$_2$O$_3$-(1×2) reconstructions can be elucidated by considering surface relaxation effects and the repulsive electrostatic interactions induced by polarons. To maintain the symmetry of the Ti$_2$O$_3$-(1×2) reconstruction, the distribution of the Ti$^{3+}$ ions in the first titanium layer must exhibit a high degree of local symmetry (Supplementary Fig. 2g–j). As shown in Supplementary Fig. 2i, j, the four Ti$^{3+}$ ions in the symmetric Ti$_2$O$_3$-(1×2) reconstruction have two types of $d$-like orbital characters ($d_{x^2-y^2}$ and $d_{z^2}$), with the $d_{x^2-y^2}$ orbital character centered on the six coordinated titanium sites and the $d_{z^2}$ orbital character centered on the five coordinated titanium sites. To mitigate repulsive electrostatic interactions between the first-nearest neighbors Ti$^{3+}$ ions, Ti$^{3+}$ ions with $d_{z^2}$ orbital character change from the first titanium layer to the second titanium layer, without breaking the symmetry of the distribution of the Ti$^{3+}$ ions in the first titanium layer (Fig. 2a and Supplementary Fig. 2g). This results in a release of surface energy by approximately 0.28 eV. An alternative way to reduce repulsive interactions involves the changing of Ti$^{3+}$ ions from the first-nearest-neighbor titanium site to the second-nearest-neighbor titanium site within the first titanium layer, breaking the local symmetry and forming the asymmetric Ti$_2$O$_3$-(1×2) reconstruction (Fig. 2b and Supplementary Fig. 2a). Energetically, this latter

mechanism proves to be more favorable by around 0.31 eV. The driving force for adopting such an asymmetric configuration is the better ability of the system to relax lattice strain induced by the more spacious Ti$^{3+}$ ion (compared to its Ti$^{4+}$ counterpart) when it is in the first titanium layer rather than in the second titanium layer. The average distance between the Ti$^{3+}$ ion and the nearest-neighbor oxygen atoms is 2.08 Å when Ti$^{3+}$ ion is located in the first titanium layer and 2.06 Å in the second titanium layer.

Furthermore, we found that the surface electronic states are sensitive to the symmetry of the Ti$_2$O$_3$-(1×2) reconstructions, influenced by surface relaxation effects and the repulsive electrostatic interactions induced by the polarons. As shown in Fig. 3, the occupied 3$d$ orbitals of the Ti$^{3+}$ ions are located near the Fermi level between the occupied 2$p$ orbitals of O atoms and the unoccupied 3$d$ orbitals of Ti. The occupied 3$d$ orbitals of the Ti$^{3+}$ ions, for both symmetric and asymmetric Ti$_2$O$_3$-(1×2) reconstructions, exhibit multiple peaks due to the different coordination environments of the Ti$^{3+}$ sites. For the symmetric Ti$_2$O$_3$-(1×2) reconstruction (Fig. 3a), the majority of the occupied 3$d$ orbitals are located at relatively high-energy levels. In contrast for the asymmetric Ti$_2$O$_3$-(1×2) reconstruction (Fig. 3b), the majority of the occupied 3$d$ orbitals are situated at relatively low-energy levels. This implies that the center of the occupied 3$d$ orbitals for the asymmetric Ti$_2$O$_3$-(1×2) reconstruction is at lower energy than that of the symmetric Ti$_2$O$_3$-(1×2) reconstruction, providing an explanation for the higher stability of the asymmetric Ti$_2$O$_3$-(1×2) reconstruction. Additionally, to determine the valence state of the top-layer atoms of the Ti$_2$O$_3$-(1×2) reconstructions, we have conducted an analysis of Bader charges. The Bader charge distributions for the top layers are presented in Supplementary Fig. 5, revealing a symmetrical charge distribution in the symmetric model and noticeable differences in the charge distribution of the asymmetric model (Supplementary Fig. 5). Both the band gap and the work function of the asymmetric Ti$_2$O$_3$-(1×2) reconstruction were found to be smaller than those of the symmetric Ti$_2$O$_3$-(1×2) reconstruction (Supplementary Fig. 6), indicating that the asymmetric Ti$_2$O$_3$-(1×2) reconstruction possesses distinct properties compared to the symmetric Ti$_2$O$_3$-(1×2) reconstruction.

### Dynamic evolution processes of rutile TiO$_2$ (110) surface

To gain further insight into how the asymmetric surface reconstruction of rutile TiO$_2$ (110)-(1×2) responds to different gas environments, we investigated its dynamic evolution under an oxygen gas environment. The stages of this evolution for the (1×2) reconstructed rutile TiO$_2$ (110) surface are depicted in Fig. 4a–f. Initially, the (1×2) reconstruction was in situ fabricated on the (110) surface at 700 °C under a vacuum environment of $6.00 \times 10^{-4}$ Pa. Upon introducing oxygen gas with a pressure of $6.00 \times 10^{-2}$ Pa, this reconstruction becomes unstable. In situ HRTEM images illustrate the dynamic change of a two-row structure (Fig. 4a–f). During this process, the distance between the two TiO$_x$ rows expands, leading to the emergence of a new adjacent TiO$_x$ row (Fig. 4e, f). This evolving pattern is reflected in the intensity profiles of the reconstructed layer, as shown in Fig. 4g. Notably, the pronounced contrast at the reconstruction sites transitions to a more uniform contrast, facilitating the analysis of intermediate states during the reconstruction. These findings show that the TiO$_2$ (110)-(1×2) is more stable under low oxygen chemical potential conditions, while the unreconstructed surface is more dominant at high oxygen chemical potentials. This provides a method to modulate these structures by adjusting the oxygen chemical potential. It is important to note that the asymmetric Ti$_2$O$_3$-(1×2) reconstruction formed at elevated temperatures remains stable even after cooling (Supplementary Fig. 7). This observation indicates that transitioning from the Ti$_2$O$_3$-(1×2) reconstruction back to the pristine surface involves crossing a certain energy barrier.

In conclusion, we have provided a comprehensive elucidation of the asymmetric surface reconstruction of rutile TiO$_2$ (110)-(1×2) through a combination of scanning transmission electron microscopy and density functional theory calculations. This reconstruction arises primarily from surface relaxation effects, coupled with the repulsive electrostatic interactions induced by the polarons. The intricate interplay between geometric and electronic structures results in distinct properties in the asymmetric Ti$_2$O$_3$-(1×2) reconstruction as compared to its symmetric counterpart. Furthermore, the observed oxygen sensitivity of the asymmetric Ti$_2$O$_3$-(1×2) reconstruction emphasizes its inherent dynamism. These findings significantly contribute to our understanding of the structural and electronic properties of reconstructed TiO$_2$ surfaces, offering valuable insights for optimizing the performance of titanium dioxide-based materials, particularly under reducing conditions.

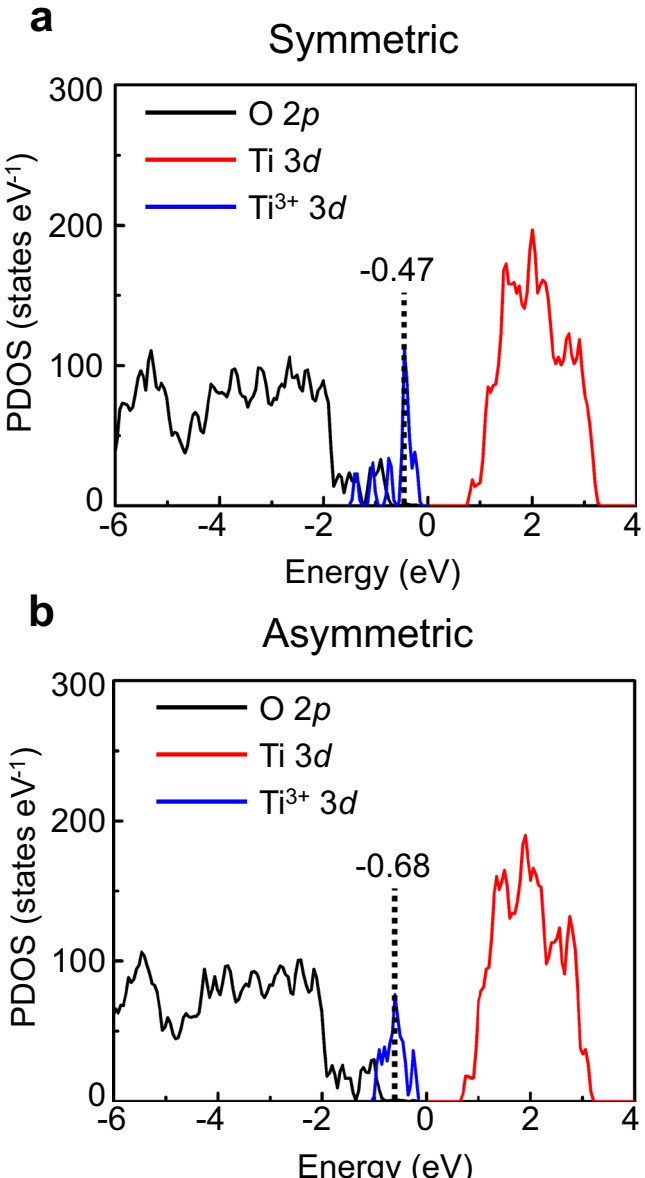

**Fig. 3 | Orbital-projected density of states of the Ti$_2$O$_3$-(1×2) reconstructions.** The occupied O 2$p$ orbitals of the entire system, unoccupied Ti 3$d$ orbitals of the entire system (Ti$^{4+}$ and Ti$^{3+}$), and the occupied Ti$^{3+}$ 3$d$ orbitals of all the Ti$^{3+}$ ions in the entire system for the symmetric (**a**) and asymmetric (**b**) Ti$_2$O$_3$-(1×2) reconstructions are displayed by black, red, and blue lines, respectively. Energies are relative to the Fermi energy level. The vertical dashed line indicates the position of the first moment of the projected $d$-band density of occupied states onto the Ti$^{3+}$ sites of the entire system.

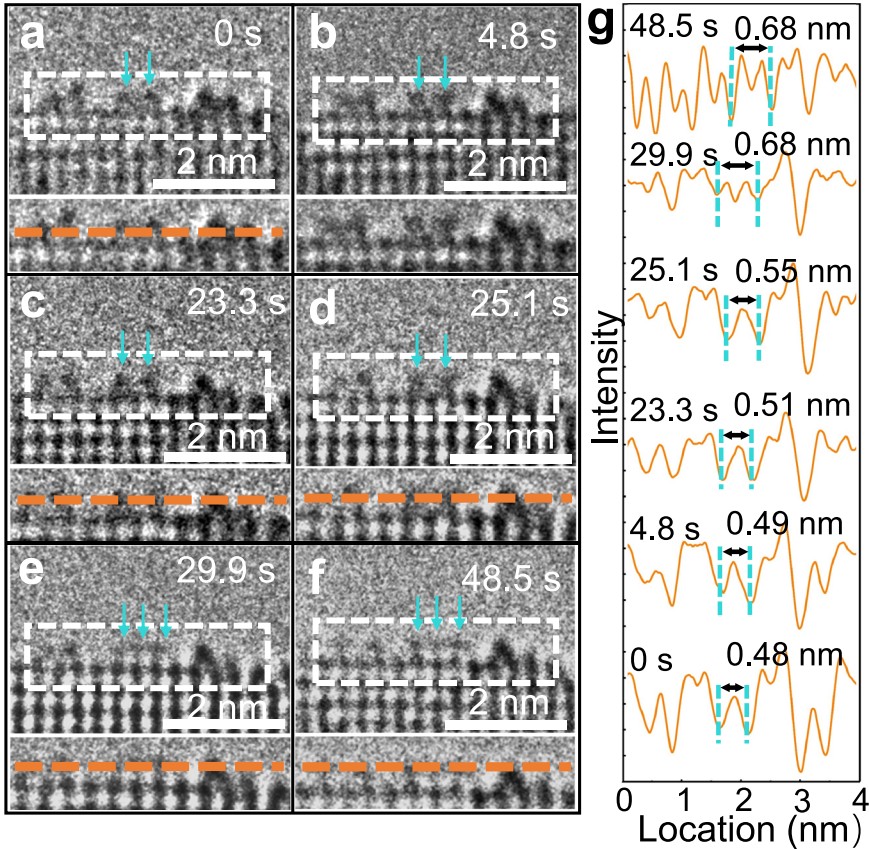

**Fig. 4 | The dynamic evolution processes of rutile TiO₂ (110) surface.**
**a–f** Sequential HRTEM images of the rutile TiO₂ (110) surface during the reconstruction under oxygen environment (pressure: $6.00 \times 10^{-2}$ Pa), acquired at 0, 4.8, 23.3, 25.1, 29.9, and 48.5 s; the rows of $TiO_x$ are indicated by the green arrows. The enlarged images of the dotted rectangles are shown in the lower panels of **a–f**, respectively. **g** Intensity profiles along the orange dashed lines in the lower panels of **a–f**. The orange dash lines are acquired from the reconstructed layer.

## Methods

### Preparation of rutile TiO₂ nanorods
In a typical synthesis process, rutile TiO₂ nanorods with (110) surface exposed are synthesized with the hydrothermal method[40]. A reaction solution is prepared by dissolving 0.015 M titanium tetrachloride (TiCl₄) directly into 3.5 M hydrochloric acid (HCl) under strong stirring in an ice water bath. Subsequently, 60 mL of the reaction solution is transferred to a Teflon-lined autoclave, followed by the addition of 0.015 M sodium fluoride (NaF). The hydrothermal reaction is then conducted at 220 °C for 12 h in an oven. Upon completion of the reaction, the products are collected through centrifugal separation, followed by washing with deionized water 4-5 times.

### In situ STEM experiments
The in situ STEM experiments were performed in a spherical aberration (Cs-) corrected scanning transmission electron microscope (200 kV, FEI Titan G2 80–200), which can provide a high spatial resolution (0.8 Å). To acquire high-quality of STEM images, all the low-order aberrations have been tuned to an acceptable level, e.g., Cₛ < 0.5 µm, A1 (2-fold astigmatism) <2 nm, A2 (3-fold astigmatism) <20 nm, and B2 (coma) <20 nm. The convergence angle used in STEM imaging was ≈21 mrad. The annular detection angle of high-angle annular dark field (HAADF) image is set to be ≈53–200 mrad, and the detection angle of a bright-field (BF) image is set to be ≈ 0–20 mrad. During the in situ STEM experiments, the as-prepared TiO₂ nanorods are dispersed into ethanol and dropped on the heating chips, which are then moved into the TEM chamber through a double tilt heating holder (Wildfire D6, DENS solutions) with a heating rate of ≈5 °C s⁻¹. Apart from directly heating our sample to 800 °C, in the step-by-step heating process, we set the temperature interval value to 100 °C. To minimize the electron beam (e-beam) damage, the e-beam illumination of the target area had only been opened during imaging[33].

### In situ ETEM experiments
ETEM experiments are performed in a Hitachi H-9500 environmental transmission electron microscope (300 kV), with a spatial resolution of 2 Å. A chipnova double tilt heating holder (CNT-SHBO-D) is used in our experiments. In a typical in situ heating experiment, the sample is heated to 700 °C with a heating rate of ≈5 °C s⁻¹, followed by the introduction of oxygen (0.06 Pa). And the gun valve is opened only when carrying images.

### Spin-polarized DFT calculations
Spin-polarized DFT calculations were carried out using the generalized gradient approximation (GGA) of Perdew-Burke-Ernzerhof (PBE) as implemented in the VASP code[41,42]. The DFT + U methodology[43,44] with an effective U value of 4.1 eV was used to describe the localized Ti 3d states, which is within the range of suitable values to describe reduced TiO₂-based systems[12,36,45]. We used projector-augmented wave (PAW) potential with Ti (3d, 4 s) and O (2 s, 2p) electrons as valence states and a plane-wave cutoff of 400 eV. The unreconstructed TiO₂ (110) surface was modeled by p(2×2) unit cells and eighteen atomic layers separated by 18 Å vacuum space to avoid interaction between periodic images. Due to the large supercell dimensions, the k-point sampling was restricted to the 2 × 1 × 1 grids. In all geometry optimizations, all atoms of the 3 bottom atomic layers were kept fixed in their bulk-truncated positions, whereas the rest of the atoms were allowed to fully relax.

The force convergence criterion for the geometry optimization was set at 0.02 eV $Å^{-1}$. For the $Ti_2O_3$-(1×2) reconstruction model, we found that the ferromagnetic state is slightly more stable than the anti-ferromagnetic state, although the difference in their stability is marginal. Consequently, we focused exclusively on the FM state for the $Ti_2O_3$-(1×2) reconstruction model in our work. The vibrational frequencies of the top layers of the $Ti_2O_3$-(1×2) reconstruction were calculated using the finite displacement method. Moreover, to confirm that our findings were not artifacts of chosen computational settings, selected systems were validated employing a larger $k$-mesh ($4 \times 2 \times 1$ $k$-points) or a larger cutoff of 500 eV or different U values, see Supplementary Table 2. These ensure the robustness of our models.

## Reporting summary

Further information on research design is available in the Nature Portfolio Reporting Summary linked to this article.

## Data availability

The data that support the findings of this study are available from the corresponding authors upon request.

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

## Acknowledgements

We acknowledge the financial support of the National Natural Science Foundation of China (52025011, 51971202, 92045301, and 52171019), the National Key Research and Development Program (2022YFA1505500), the Key Research and Development Program of Zhejiang Province (2021C01003), the Zhejiang Provincial Natural Science Foundation of China (LR23B030004, LD19B030001), Shanxi-Zheda Institute of Advanced Materials and Chemical Engineering and the Fundamental Research Funds for the Central Universities. M.V.G.P. thanks for grant PID2021-128915NB-I00 funded by MCIN/AEI/ 10.13039/501100011033 and by the European Regional Development Fund - "A way of making Europe". The authors thank Dr. Zhemin Wu in Center of Electron Microscopy at Zhejiang University for TEM data analysis.

## Author contributions

Y.W. designed and supervised the project. M.V.G.P. and Z.-K.H. supervised the density functional calculations. W.T.Y. and B.W.C. performed the STEM experiments. R.Y.Y. conducted structural simulations. Y.J. and G.X.L. participated in data analysis and discussion. R.Q. and Z.-K.H. performed the calculations. H.L.W. prepared rutile samples. Z.-K.H., W.T.Y., and B.W.C. wrote the manuscript with inputs from all the authors. All authors contributed to the analysis and interpretation of the results. All the authors commented on the manuscript and have given approval to the final version of the manuscript.

## Competing interests

The authors declare no competing interests.
