## [Peer Review File · Nature Communications]

Direct in-situ insights into the asymmetric surface reconstruction of rutile TiO₂ (110)REVIEWER COMMENTS

Reviewer #1 (Remarks to the Author):

Overall I find that this paper is a nice piece of work that is appropriate to publish. However, I do have two significant changes that I will require that the authors make:

1) This is a reduced surface. The authors never clearly state this, they have to in the title. I suggest a change to:

"Direct in Situ Determination of a Reduced, Asymmetric Surface Reconstruction on Rutile TiO₂ (110)"

2) They must remove the term "small polaron" as this obfuscates. In normal solid-state usage small polarons are quasiparticles combining charge and local distortions which play a role in charge transport. Here these are immobile, reduced Ti³⁺ atoms. The use of the term "small polaron" both confuses and makes the paper appear to be anomalously novel.

Reviewer #2 (Remarks to the Author):

In this manuscript, the authors study the atomic structure of rutile TiO₂ (110) surface (1×2) reconstruction combining in situ spherical aberration-corrected STEM and DFT calculations. Their findings confirmed the asymmetric Ti₂O₃-(1×2) reconstruction model under high vacuum and high temperature environment. And they unraveled that the formation mechanism of such reconstruction is polaronic effects. In addition, the emergence and dissolution of these asymmetric structures can be modulated by tuning the oxygen partial pressure. These results are of great interest to the scientists in oxide surfaces and catalysis. I recommend publication after revision.

Below are some suggestions/questions to the authors.

1) For the Fig. 1b (BF-STEM image), due to resolution limitations, the oxygen atoms at the top layer are not obvious. The experimental atomic structure may be the Ti₂O-(1×2) model proposed by Park et al. (Ref. 24). Therefore, for the DFT calculations, the Ti₂O-(1×2) model should also be considered. The Ti₂O₃-(1×2) reconstruction model must be further confirmed, for example, by comparing the atomic positions and atomic spacings of surface layers of different models.

2) Details about the calculation method of the small polarons/Ti³⁺ ions and explanation on the polaronic effects are lacking. In Figure S2, there are various distributions of the Ti³⁺ ions. Are these Ti³⁺ distribution designed intentionally? Some Ti³⁺ is deep in the bulk, why?

3) Bader charge analysis or bond valence sum (BVS) should be performed to obtain the valence state of the top-layer atoms.

4) There seem some contradictions about the electronic structure of the surface models and their stability. For example, the low-energy states (e.g., those near -1.5 eV) of the Ti³⁺ ions in the symmetric model are absent in the asymmetric model, although for the latter the peak energy is lower.

5) Smaller band gap and the work function of the asymmetric reconstruction also suggest that it is energetically less stable than the symmetric one.

6) "At the beginning of the reconstruction, three TiO_x double-rows can be observed within the dotted rectangles, presenting a 3d-3d pattern." This is interesting. Is it an incidental phenomenon or a general result? How many times has the reconstruction model (3d-3d pattern) observed? Is it a new stable reconstruction? More detailed explanation is needed.

7) Generally, the atomic structures of the rutile TiO₂ (110) surface under different oxygen partial pressure at room temperature are also very important. Is it the TiO₂(110)-(1×1) structure discussed in the article? When it comes down to room temperature, is the Ti₂O₃-(1×2) reconstruction stable at room temperature? In this work, the authors did not mention it. If the experiment has been done, the authors should discuss it.

8) The DFT simulation scheme needs to be described in detail. How the authors deal with the spin structure in the DFT calculations? The ideal rutile TiO₂ is diamagnetic in nature, but room temperature ferromagnetic can be induced by oxygen vacancies. For the Ti₂O₃-(1×2) reconstruction model, the oxygen vacancies exist at the surface, the FM or AFM ground states should be predicted, and the

magnetic moments of surface atoms should be given.

The following are the minor comments.

- 1) Supplementary Figures 1d and 1e mentioned in the manuscript and supplementary material are not included.
- 2) Surface free energy as a function of oxygen pressure and temperature is lacking.
- 3) Details about the vibrational entropy correction are lacking.
- 4) "Several structural models have been proposed to elucidate the (1×2) reconstruction of the rutile (110) surface." Do these models presented refer to the supplementary Figure 2? What's the difference? Please provide more detailed explanation.
- 5) For the plan-view models, the atoms of the different layers are not clearly distinguishable. Please modify these pictures further.
- 6) In "(Figure 2b and Supplementary Figure 2g)", Figure 2b should be Figure 2a. In "(Figure 2a and Supplementary Figure 2a)", Figure 2a should be Figure 2b.
- 7) "Conversely, for the asymmetric TiO₂-(1×2) reconstruction (Figure 3b) minority of the occupied 3d orbitals are located at high energy levels." High energy levels should be "low energy levels".
- 8) The Fermi energy should be labelled in Figure 3.

Reviewer #3 (Remarks to the Author):

Reviewer #4 (Remarks to the Author):

The work of W. Yuan and coworkers reports a combined experimental (TEM) and first-principles calculations (DFT) to study the (1×2) reconstruction of rutile (1×2). The work is overall interesting; in particular, the authors provide a high-resolution side-view of this well-known reconstruction, providing direct (lateral) evidence of the structural characteristics of this reconstruction and explaining the degree of structural asymmetry through orbital-dependent polaron-polaron interactions. These two results are valid, scientifically robust, and sound.

Considering the long-standing interest in rutile TiO₂ and the increasing interest in the role of polarons in altering the physical properties of materials, especially on surfaces, this work might deserve publication in NatComm. However, there are aspects of the presentation that need to be modified before the paper can be considered for publication:

- 1) The authors declare in the abstract that they 'unravel the asymmetric surface reconstruction of rutile TiO₂(1×2)', which is not correct and needs to be amended. The symmetric character of the surface has been reported in Ref. 12 (DFT+evolutionary algorithms) and 32 (experiment). Moreover, the authors claim that there are controversy that they resolve with this study, which is also not correct. Ref. 12 and 32 have unambiguously demonstrated that rutile TiO₂(1×2) reconstructs with an asymmetric model, results that have also been confirmed in Ref. 13 by AFM and DFT-based molecular dynamics. The authors should re-write the introduction, giving the proper credit to these early works, avoiding delivering the impression that they have resolved this issue in this work.
- 2) The authors find that tuning the oxygen partial pressure is a means to control the degree of asymmetry, which is a nice and interesting specific result. However, the authors conclude that 'this research offers an innovative approach to understand better...surface reconstruction'. I am afraid I

have to disagree with this statement. What they found is specific to this surface; no results indicate a general validity of this well-controlled tuning procedure. In addition, this approach is not novel; changing the oxygen partial pressure to modify surface properties is a very general approach widely used in many surface-sensitive techniques. The authors should weaken this statement both in the abstract and conclusion.

3) Finally, as stated above, the authors interpret the structural asymmetry as due to orbital-dependent polaron-polaron interaction, which is an interesting and novel result. Also, in this case, as in the above two points, the reader has the impression that the authors tend to weaken the importance of previous works. The central role of polarons in driving the formation of the 1x2 reconstruction was discussed in Ref. 13, and this should be clearly stated. Based on these previous results, the authors could build a physical model based on orbital-dependent polaron. The importance of the polaron-orbital on the polaron-polaron interaction and energy was introduced in Phys. Rev. B 98, 045306 (2018) and <https://doi.org/10.1038/s41524-022-00805-8>. The authors should modify the text accordingly.

In summary, this work's main novelty and exciting aspects are the high-quality side-view TEM image and the interpretation of the structural asymmetry due to polaron-polaron interaction. Other claims/results, commented in points 1, 2 and 3, are less important and sometimes misleading. After proper revision, the paper can be recommended for publication in NatCom.

Reviewer #5 (Remarks to the Author):

This article contains in-situ spherical aberration corrected scanning TEM (STEM) images that are at the "state of the art" as well as DFT calculations. The images are really outstanding and suggest, even by a visual inspection, the emergence of a Ti₂O₃ stoichiometry-based surface reconstruction. Interestingly, the authors find that differently to reported in previous structural determinations by electron diffraction and STM, an asymmetrical configuration of the Ti atoms within the Ti₂O₃ model would be favorable.

However, this article could it be very important 10 years ago. Indeed, the most important papers cited in the article correspond to the period between 2006 and 2009. In this period structure of surface reconstruction was a trending topic as they appeared as a way to control surface reactivity and electronic properties. However, 15 years later, surface reconstructions that are stable in high and ultra high vacuum are not considered as a technologically valuable material. Therefore, the technological relevance of the findings of the manuscript is very limited. (I question the last sentence: ... pave the way for fine-tuning surface reconstructions, which have important implications in the optimization of the performance of titanium dioxide-based devices).

This is the reason why I do not recommend this article for publication in Nat. Comm., a journal with a scope of interdisciplinary research with general implications, but in a more focused journal.

Some particular points:

The Ti₂O₃ asymmetrical reconstruction is only seen in some of the rows of the fig. 1 STEM images. Some "dimers" looks symmetrical whereas other looks asymmetrical. Authors have not discussed this point. That means that both reconstructions are stable?

Calculations are performed at T = 0, however, it could be possible that at RT, Ti atoms would be oscillating from one position to the other in the asymmetrical model. Could you discuss this process?

In this case, how the STEM image would look like

Could this be caused by knock-on of the electron irradiation?

Minor point: Many of the references lacks of the page, and they have been difficult to find.

Reviewer #6 (Remarks to the Author):

Response to Reviewers

Reviewer 1

General comment: Overall, I find that this paper is a nice piece of work that is appropriate to publish. However, I do have two significant changes that I will require that the authors make:

Response: Thank you for the positive feedback. In response to your comments, we have made the necessary revisions to the manuscript and believe we have addressed all your concerns.

- 1) This is a reduced surface. The authors never cleanly state this, they have to in the title. I suggest a change to: “Direct in Situ Determination of a Reduced, Asymmetric Surface Reconstruction on Rutile TiO₂ (110)”

Response: Thank you for your valuable feedback. Following your suggestions, we have updated the title to “Direct *in Situ* Determination of a Reduced, Asymmetric Surface Reconstruction on Rutile TiO₂ (110)”.

- 2) They must remove the term “small polaron” as this obfuscates. In normal solid-state usage small polarons are quasiparticles combining charge and local distortions which play a role in charge transport. Here these are immobile, reduced Ti³⁺ atoms. The use of the term “small polaron” both confuses and makes the paper appear to be anomalously novel.

Response: Thank you for your input. We agree that polarons constitute quasiparticles, emerging from the interactions between electrons and the crystal lattice. This phenomenon becomes significant in the context of oxygen-deficient reducible oxides such as rutile TiO₂. The absence of oxygen atoms in the lattice results in excess electrons, which tend to localize at cationic sites near the oxygen vacancies rather than being uniformly distributed across the crystal, driving the Ti⁴⁺ (*d*⁰) to Ti³⁺ (*d*¹) reduction. This localization induces a local distortion in the positions of the nearby atoms to the Ti³⁺ site, effectively creating a “bubble” around the electron, recognize as the small polaron. While the term “small polaron” is commonly used in the literature as equivalent to Ti³⁺, we chose to use it in our manuscript for consistency, without implying the discovery of any novel finding. [see Reticcioli, M. *et al.* Polaron-driven surface reconstructions. *Physical Review X* **7**, 031053 (2017); Reticcioli, M. *et al.* Formation and dynamics of small polarons on the rutile TiO₂(110) surface. *Physical Review B* **98**, 045306 (2018); Birschitzky, V. *et al.* Machine learning for exploring small polaron configurational space. *npj Computational Materials* **8**, 125 (2022).]

Reviewer 2

General comment: In this manuscript, the authors study the atomic structure of rutile TiO_2 (110) surface (1×2) reconstruction combining in situ spherical aberration-corrected STEM and DFT calculations. Their findings confirmed the asymmetric Ti_2O_3 -(1×2) reconstruction model under high vacuum and high temperature environment. And they unraveled that the formation mechanism of such reconstruction is polaronic effects. In addition, the emergence and dissolution of these asymmetric structures can be modulated by tuning the oxygen partial pressure. These results are of great interest to the scientists in oxide surfaces and catalysis. I recommend publication after revision. Below are some suggestions/questions to the authors.

Response: Thank you for the positive and insightful feedback. It has been instrumental in enhancing the quality of our work. In response to your comments, we have made the necessary revisions to our manuscript, and are confident that we have fully addressed all your concerns.

1) For the Fig. 1b (BF-STEM image), due to resolution limitations, the oxygen atoms at the top layer are not obvious. The experimental atomic structure may be the Ti_2O -(1×2) model proposed by Park *et al.* (Ref. 24). Therefore, for the DFT calculations, the Ti_2O -(1×2) model should also be considered. The Ti_2O_3 -(1×2) reconstruction model must be further confirmed, for example, by comparing the atomic positions and atomic spacings of surface layers of different models.

Response: In response of the valuable feedback provided by the reviewer, we have performed calculations on the Ti_2O -(1×2) model as proposed by Park *et al.* Various configurations of Ti^{3+} were systematically explored to identify the most stable Ti^{3+} distribution, as depicted in Figure R1. To assess the relative stability between the Ti_2O -(1×2) model and the Ti_2O_3 -(1×2) reconstruction model, we generated a phase diagram plotting surface formation energies per unit cell for both models as a function of the oxygen chemical potential, relative to the unreconstructed surface (Figure R2). The phase diagram reveals a transition in the TiO_2 (110) surface structure as the oxygen chemical potential decreases or, equivalently, at constant oxygen partial pressure with increasing temperature. Specifically, the surface evolves from the unreconstructed state to the Ti_2O_3 -(1×2) reconstruction model, consistent with earlier theoretical studies.^{12,24} Under our experimental conditions, the Ti_2O_3 -(1×2) reconstruction model proves to be more stable than the Ti_2O -(1×2) model. Moreover, our detailed calculations reveal an inherent symmetry in the Ti_2O -(1×2) model is inherently symmetric, with any starting asymmetric configuration proving to be unstable and reverting to a symmetric form after first-principles structural optimization. Consequently, the asymmetric configuration observed in our experiments does not align with the Ti_2O -(1×2) model but it is representative of the Ti_2O_3 -(1×2) reconstruction model.

Changes made:

1. On page 4 the text has been modified:

“...although the oxygen atoms in the outermost layer were not distinctly visible in our experiments due to resolution constraints. To assess whether the observed atomic structure aligns with the Ti_2O -(1×2) model proposed by Park et al.,²⁴ we performed calculations on this model. Various configurations of Ti^{3+} were considered for determining the most stable Ti^{3+} distribution, as depicted in Supplementary Figure 3. Phase diagrams plotting surface formation energies per unit cell for both the Ti_2O_3 -(1×2) reconstruction model and the Ti_2O -(1×2) model under varying oxygen chemical potentials, relative to the unreconstructed surface, are presented in Supplementary Figure 4. These diagrams reveal that with decreasing oxygen chemical potential (or at constant oxygen partial pressure with increasing temperature), the TiO_2 (110) surface undergoes a transition from the unreconstructed surface to the Ti_2O_3 -(1×2) reconstruction model, consistent with earlier theoretical studies.^{12,24} Under our experimental conditions, the Ti_2O_3 -(1×2) reconstruction model proves more stable than the Ti_2O -(1×2) model. Additionally, our detailed calculations reveal that the Ti_2O -(1×2) model is inherently symmetric, with its asymmetric configuration being unstable and reverting to a symmetric form following first-principles structural optimization. Consequently, the asymmetric configuration observed in our experiments is not representative of the Ti_2O -(1×2) model, but rather of the Ti_2O_3 -(1×2) reconstruction model.”

2. We have added Figure R1 and R2 into the revised Supplementary Materials.

Figure R1. Multiple distributions of the polarons in the near-surface of the Ti_2O - (1×2) reconstruction. The titanium and oxygen atoms are represented by white and red balls, respectively. The energy values are relative to the most stable configuration.

Figure R2. Surface phase diagram of rutile TiO_2 (110): stability of different structures as a function of oxygen chemical potential $\Delta\mu$.

- 2) Details about the calculation method of the small polarons/ Ti^{3+} ions and explanation on the polaronic effects are lacking. In Supplementary Figure 2, there are various distributions of the Ti^{3+} ions. Are these Ti^{3+} distribution designed intentionally? Some Ti^{3+} is deep in the bulk, why?

Response: Thank you for bringing up these concerns. The absence of oxygen atoms in the lattice results in excess electrons, which tend to localize at cationic sites near the oxygen vacancies rather than being uniformly distributed across the crystal, driving the Ti^{4+} (d^0) to Ti^{3+} (d^1) reduction. This localization induces a local distortion in the positions of the nearby atoms to the Ti^{3+} site, effectively creating a “bubble” around the electron, recognize as the small polaron. It’s important to note that polarons in reducible oxides do not necessarily position themselves adjacent to vacancy sites [Phys. Rev. Lett. 102, 026101 (2009); Phys. Rev. B 79, 193401 (2009); J. Phys. Chem. C 115, 15, 7562–7572 (2011)]; they can be distributed in various ways. In determining the most stable distribution of the excess charge in the Ti_2O_3 -(1x2) reconstructed surface, we explored multiple polaron distributions, including some with Ti^{3+} ions in deeper layers. The most

stable configuration, as identified, is shown in Figure 2, with all considered configurations presented in Supplementary Figure 2. Since polarons carry identical charges, they inherently repel each other electrostatically, leading to a natural tendency to maintain distance from one another. Moreover, surface relaxation effects, as discussed in existing literature [Phys. Rev. Lett. 102, 026101 (2009)], play a crucial role in influencing their distribution. These effects tend to drive polarons towards surface areas rather than deeper into the bulk.

3) Bader charge analysis or bond valence sum (BVS) should be performed to obtain the valence state of the top-layer atoms.

Response: Thank you for your suggestion. Following your advice, we have conducted an analysis of Bader charges. The Bader charge distributions for the top layers are compiled in Figure R3. As observed from Figure R3, it is evident that the charge distribution in the symmetric model is symmetrical, while noticeable differences exist in the charge distribution of the asymmetric model.

Figure R3. The Bader charge distribution in the top layers of the $\text{Ti}_2\text{O}_3-(1 \times 2)$

reconstructions.

Changes made:

1. On pages 6 the text has been modified:

“Additionally, to determine the valence state of the top-layer atoms of the Ti_2O_3 -(1×2) reconstructions, we have conducted an analysis of Bader charges. The Bader charge distributions for the top layers are presented in Supplementary Figure 5, revealing a symmetrical charge distribution in the symmetric model and noticeable differences in the charge distribution of the asymmetric model (Supplementary Figure 5).”

2. We have added Figure R3 into the revised Supplementary Materials.

4) There seem some contradictions about the electronic structure of the surface models and their stability. For example, the low-energy states (e.g., those near -1.5 eV) of the Ti^{3+} ions in the symmetric model are absent in the asymmetric model, although for the latter the peak energy is lower.

Response: We greatly appreciate the reviewer’s meticulous examination of the density of states (DOS) diagrams. To provide a more comprehensive understanding of the contribution of *d*-electrons to stability, we have calculated the first moment of the projected *d*-band density of occupied states onto the Ti^{3+} sites of the entire system. For the asymmetric Ti_2O_3 -(1×2) reconstruction, the center of gravity of the occupied 3*d* orbitals lies at a lower energy compared to that of the symmetric Ti_2O_3 -(1×2) reconstruction (-0.68 and -0.47 eV, respectively). This result supports the conclusion that the asymmetric Ti_2O_3 -(1×2) reconstruction exhibits higher stability.

5) Smaller band gap and the work function of the asymmetric reconstruction also suggest that it is energetically less stable than the symmetric one.

Response: We appreciate the reviewer for highlighting this issue. Band gaps can serve as a reliable indicator of stability for perfectly ionic and non-polar surfaces. In such cases, it is commonly agreed that a larger band gap implies better stabilization of filled electronic states, leading to more effective energy minimization. However, surface Gibbs energies are influenced by numerous other factors, none of which directly correlate with the band gap. For instance:

- Polar surface arrangements significantly increase surface energy but may not impact the band gap.
- Dangling bond states increase surface energy without reducing the band gap, except when defect states are positioned directly within the gap.
- Surface energies of oxides are contingent on oxygen chemical potentials, a factor not reflected in the band gap.

Therefore, the observation that a material with a lower band gap exhibits higher thermodynamic stability is not contradictory and aligns with our research findings.

- 6) “At the beginning of the reconstruction, three TiO_x double-rows can be observed within the dotted rectangles, presenting a 3d-3d pattern.” This is interesting. Is it an incidental phenomenon or a general result? How many times has the reconstruction model (3d-3d pattern) observed? Is it a new stable reconstruction? More detailed explanation is needed.

Response: Thank you for the insightful comment. Based on our *in situ* experiments, we can affirm that this phenomenon is not a random occurrence. During the reconstruction process in vacuum, multiple TiO_x double-rows became apparent on the rutile TiO_2 (110) surface, exhibiting intervals of varying distances, predominantly 1d, 2d, and 3d. Notably, as the reconstruction progressed, the 2d and 3d patterns consistently transformed into 1d patterns. This consistent transformation suggests that the 3d-3d pattern represents a metastable structure that emerges in the early phase of the reconstruction process.

- 7) Generally, the atomic structures of the rutile TiO_2 (110) surface under different oxygen partial pressure at room temperature are also very important. Is it the TiO_2 (110)-(1×1) structure discussed in the article? When it comes down to room temperature, is the Ti_2O_3 -(1×2) reconstruction stable at room temperature? In this work, the authors did not mention it. If the experiment has been done, the authors should discuss it.

Response: We appreciate the insightful query from the reviewer. As depicted in the phase diagram (Figure R2), thermodynamic analysis suggests that at room temperature, the pristine, unreconstructed surface exhibits greater stability compared to the Ti_2O_3 -(1×2) reconstruction. To comprehensively address this point, we have conducted supplementary experiments specifically examining the stability of the Ti_2O_3 -(1×2) reconstruction during the cooling process. Our latest experimental findings, shown in Figure R4, reveal that the Ti_2O_3 -(1×2) reconstruction can indeed persist at room temperature. This implies that, while the pristine surface may be more stable from a thermodynamic standpoint, the Ti_2O_3 -(1×2) reconstruction formed at elevated temperatures remains stable even after cooling. This observation indicates that transitioning from the Ti_2O_3 -(1×2) reconstruction back to the pristine surface involves crossing a certain energy barrier.

Figure R4. The atomic-resolution HAADF STEM images of the (1×2) reconstructed rutile TiO₂ (110) surface. (a-b) Sequential in situ HADDF images of TiO₂ (110) surface at (a) 900 °C and (b) 20 °C. These images are acquired *in situ* in a vacuum ($\sim 10^{-5}$ Pa).

Changes made:

1. On page 8 the text has been modified:
 “It is important to note that the asymmetric Ti₂O₃-(1×2) reconstruction formed at elevated temperatures remains stable even after cooling (Supplementary Figure 7). This observation indicates that transitioning from the Ti₂O₃-(1×2) reconstruction back to the pristine surface involves crossing a certain energy barrier.”
 2. We have added Figure R4 into the revised Supplementary Materials.
- 8) The DFT simulation scheme needs to be described in detail. How the authors deal with the spin structure in the DFT calculations? The ideal rutile TiO₂ is diamagnetic in nature, but room temperature ferromagnetic can be induced by oxygen vacancies. For the Ti₂O₃-(1×2) reconstruction model, the oxygen vacancies exist at the surface, the FM or AFM ground states should be predicted, and the magnetic moments of surface atoms should be given.

Response: We appreciate insightful query from the reviewer. In this study, all calculations were spin-polarized calculations. Specifically, for the Ti₂O₃-(1×2) reconstruction model, we found that the ferromagnetic (FM) state is slightly more stable than the antiferromagnetic (AFM) state, although the difference in their stability is marginal. Consequently, we focused exclusively on the FM state for the Ti₂O₃-(1×2) reconstruction model in our work. To enhance clarity and completeness, further details on this aspect have now been added to the methodology section of the revised manuscript for clarity and completeness.

9) The following are the minor comments.

1. Supplementary Figures 1d and 1e mentioned in the manuscript and supplementary

material are not included.

Response: The corresponding images have been added to the updated Supplementary Figure 1.

2. Surface free energy as a function of oxygen pressure and temperature is lacking.

Response: In the updated Supplementary Materials, we have included the corresponding phase diagrams.

3. Details about the vibrational entropy correction are lacking.

Response: Details about the vibrational entropy correction have been added to the methodology section of the updated manuscript.

4. “Several structural models have been proposed to elucidate the (1×2) reconstruction of the rutile (110) surface.” Do these models presented refer to the supplementary Figure 2? What’s the difference? Please provide more detailed explanation.

Response: In Supplementary Figure 2, various configurations of Ti^{3+} ions are considered for the Ti_2O_3 -(1×2) reconstruction models consider in order to identify the most stable Ti^{3+} configuration.

5. For the plan-view models, the atoms of the different layers are not clearly distinguishable. Please modify these pictures further.

Response: We have now provided top and side views of each configuration, along with the coordinate information for these configurations in the Supplementary Materials. This addition aims to enhance the convenience and clarity of our presentation for our readers.

6. In “(Figure 2b and Supplementary Figure 2g)”, Figure 2b should be Figure 2a. In “(Figure 2a and Supplementary Figure 2a)”, Figure 2a should be Figure 2b.

Response: We have made the necessary revisions.

7. “Conversely, for the asymmetric Ti_2O_3 -(1×2) reconstruction (Figure 3b) minority of the occupied 3d orbitals are located at high energy levels.” High energy levels should be “low energy levels”.

Response: We have made the necessary revisions.

8. The Fermi energy should be labelled in Figure 3.

Response: In response to your suggestion, we have adjusted Figure 3 accordingly.

Reviewer 3

General comment:

Reviewer 4

General comment:

The work of W. Yuan and coworkers reports a combined experimental (TEM) and first-principles calculations (DFT) to study the (1×2) reconstruction of rutile (1×2). The work is overall interesting; in particular, the authors provide a high-resolution side-view of this well-known reconstruction, providing direct (lateral) evidence of the structural characteristics of this reconstruction and explaining the degree of structural asymmetry through orbital-dependent polaron-polaron interactions. These two results are valid, scientifically robust, and sound. Considering the long-standing interest in rutile TiO₂ and the increasing interest in the role of polarons in altering the physical properties of materials, especially on surfaces, this work might deserve publication in NatComm. However, there are aspects of the presentation that need to be modified before the paper can be considered for publication:

Response: Thank you to the reviewer for the positive and insightful comments, which have been instrumental in enhancing the quality of our work. We have made the appropriate revisions to the manuscript based on your feedback. Specifically, in the new revision, we have more clearly articulated the significant role of previous work in understanding the Ti₂O₃-(1×2) reconstruction, as well as the innovative aspects of our current study.

- 1) The authors declare in the abstract that they ‘unravel the asymmetric surface reconstruction of rutile TiO₂(1x2)’, which is not correct and needs to be amended. The symmetric character of the surface has been reported in Ref. 12 (DFT + evolutionary algorithms) and 32 (experiment). Moreover, the authors claim that there is controversy that they resolve with this study, which is also not correct. Ref. 12 and 32 have unambiguously demonstrated that rutile TiO₂(1x2) reconstructs with an asymmetric model, results that have also been confirmed in Ref. 13 by AFM and DFT-based molecular dynamics. The authors should re-write the introduction, giving the proper credit to these early works, avoiding delivering the impression that they have resolved this issue in this work.

Response: Thank you for your feedback. We have revised the manuscript in accordance with your suggestions.

Changes made:

1. On page 1 the abstract has been modified:

“The reconstruction of rutile TiO_2 (110) holds significant importance as it profoundly influence the surface chemistry and catalytic properties of this widely used material in various applications, from photocatalysis to solar energy conversion. This study employs cutting-edge, in situ spherical aberration corrected scanning transmission electron microscopy, complemented by density functional theory calculations, enabling direct observation of the asymmetric surface reconstruction of rutile TiO_2 (110)-(1 \times 2) with atomic resolution.”

2. On page 2 the text has been modified:

“In the meanwhile, the global optimization method¹² and studies using total reflection high-energy positron diffraction³² suggest the existence of an asymmetric Ti_2O_3 -(1 \times 2) reconstruction driven by polarons.¹³”

2) The authors find that tuning the oxygen partial pressure is a means to control the degree of asymmetry, which is a nice and interesting specific result. However, the authors conclude that ‘this research offers an innovative approach to understand better...surface reconstruction’. I am afraid I have to disagree with this statement. What they found is specific to this surface; no results indicate a general validity of this well-controlled tuning procedure. In addition, this approach is not novel; changing the oxygen partial pressure to modify surface properties is a very general approach widely used in many surface-sensitive techniques. The authors should weaken this statement both in the abstract and conclusion.

Response: Thank you for your feedback. We have revised the manuscript in accordance with your suggestions.

Changes made:

1. On page 1 the abstract has been modified:

“This research provides an avenue for a deeper understanding, prediction, and manipulation of the surface reconstructions of rutile TiO_2 (110), holding implications for a diverse range of applications and technological advancements involving rutile-based materials.”

2. On page 8 the text has been modified:

“These findings significantly contribute to our understanding of the structural and electronic properties of reconstructed TiO_2 surfaces, offering valuable insights for optimizing the performance of titanium dioxide-based materials, particularly under reducing conditions.”

- 3) Finally, as stated above, the authors interpret the structural asymmetry as due to orbital-dependent polaron-polaron interaction, which is an interesting and novel result. Also, in this case, as in the above two points, the reader has the impression that the authors tend to weaken the importance of previous works. The central role of polarons in driving the formation of the 1x2 reconstruction was discussed in Ref. 13, and this should be clearly stated. Based on these previous results, the authors could build a physical model based on orbital-dependent polaron. The importance of the polaron-orbital on the polaron-polaron interaction and energy was introduced in Phys. Rev. B 98, 045306 (2018) and <https://doi.org/10.1038/s41524-022-00805-8>. The authors should modify the text accordingly.

Response: Thank you for your feedback. We have revised the manuscript in accordance with your suggestions.

Changes made:

1. On page 2 the text has been modified:

“In the meanwhile, the global optimization method¹² and studies using total reflection high-energy positron diffraction³² suggest the existence of an asymmetric Ti_2O_3 -(1×2) reconstruction driven by polarons.¹³”

2. On page 5 the text has been modified:

“Building upon previous studies that highlighted the significant influence of orbital-dependent polaron-polaron interactions on system stability,^{38,39} our initial investigation focuses on the role of polaronic effects in the formation of Ti_2O_3 -(1×2) reconstructions.”

- 4) In summary, this work’s main novelty and exciting aspects are the high-quality side-view TEM image and the interpretation of the structural asymmetry due to polaron-polaron interaction. Other claims/results, commented in points 1, 2 and 3, are less important and sometimes misleading. After proper revision, the paper can be recommended for publication in NatCom.

Response: We express our gratitude once again for the positive comments provided by the reviewer and appreciate the constructive suggestions for revisions. Following these suggestions, we have made the necessary modifications and hope that the reviewer finds the new version suitable for publication in Nature Communications.

Reviewer 5

General comment:

This article contains in-situ spherical aberration corrected scanning TEM (STEM) images that are at the “state of the art” as well as DFT calculations. The images are really outstanding and suggest, even by a visual inspection, the emergency of a Ti_2O_3 stoichiometry-based surface reconstruction. Interestingly, the authors find that differently to reported in previous structural determinations by electron diffraction and STM, an asymmetrical configuration of the Ti atoms within the Ti_2O_3 model would be favorable. However, this article could it be very important 10 years ago. Indeed, the most important papers cited in the article correspond to the period between 2006 and 2009. In this period structure of surface reconstruction was a trending topic as they appeared as a way to control surface reactivity and electronic properties. However, 15 years later, surface reconstructions that are stable in high and ultra-high vacuum are not considered as a technologically valuable material. Therefore, the technological relevance of the findings of the manuscript is very limited. (I question the last sentence: ... pave the way for fine-tuning surface reconstructions, which have important implications in the optimization of the performance of titanium dioxide-based devices). This is the reason why I do not recommend this article for publication in Nat. Comm., a journal with a scope of interdisciplinary research with general implications, but in a more focused journal.

Response: We sincerely appreciate the positive assessment provided by the reviewer regarding our research on the surface reconstructions of TiO_2 . Surface reconstructions are closely linked to the physicochemical properties of materials, particularly in catalysis, where the catalytic performance is often intricately tied to the surface structure. As a material widely used in thermal and photocatalysis, the surface reconstruction of TiO_2 has always been a focal point of research. While surface reconstructions stable in high and ultra-high vacuum conditions may not initially appear technically promising, for reducible metal oxides like TiO_2 , these conditions lead to oxygen deficiency on the surface and the formation of polarons, which in turn trigger reconstructions. In numerous critical industrial applications, TiO_2 is commonly subjected to reducing conditions, such as in the hydrogenation reaction of carbon dioxide. These reducing conditions, akin to high vacuum conditions, have similar effects in inducing reconstructions. Consequently, studying surface reconstructions under high vacuum conditions can offer valuable insights for applications of TiO_2 under industrial reducing conditions. In summary, for TiO_2 , research into its surface reconstructions plays a crucial role in deepening our understanding of its applications in catalysis and further enhancing its catalytic activity.

Changes made:

On page 8 the text has been modified:

“These findings significantly contribute to our understanding of the structural and electronic properties of reconstructed TiO₂ surfaces, offering valuable insights for optimizing the performance of titanium dioxide-based materials, particularly under reducing conditions.”

- 1) The Ti₂O₃ asymmetrical reconstruction is only seen in some of the rows of the fig. 1 STEM images. Some “dimers” looks symmetrical whereas other looks asymmetrical. Authors have not discussed this point. That means that both reconstructions are stable?

Response: Thank you to the reviewer for the meticulous examination of our experimental images. Indeed, there are both symmetric and asymmetric Ti₂O₃-(1×2) reconstructions, with the asymmetric form being more stable by approximately 0.3 eV. This aligns with our *in situ* STEM observations, where most of the Ti₂O₃-(1×2) reconstructions observed are asymmetric. This point has been addressed on page 3 of the revised manuscript.

- 2) Calculations are performed at T=0, however, it could be possible that at RT, Ti atoms would be oscillating from one position to the other in the asymmetrical model. Could you discuss this process? In this case, how the STEM image would look like?

Response: Thank you for the insightful question posed by the reviewer. In the asymmetrical model, from a thermodynamic perspective, the initial and final states of Ti atoms oscillating between positions are symmetrically equivalent. In our experimental observations of the asymmetric Ti₂O₃-(1×2) reconstructions, we found that the direction in which Ti columns move, either upwards or downwards, seems to occur randomly.

- 3) Could this be caused by knock-on of the electron irradiation?

Response: Thank you to the reviewer for raising this concern. By varying the electron beam dosage, we arrived at the same conclusion. Moreover, in our experiments, we made every effort to minimize electron beam irradiation damage. Therefore, we are confident that these results are not caused by electron beam irradiation.

- 4) Minor point: Many of the references lacks the page, and they have been difficult to find.

Response:

In response to the reviewer’s comments, we have made the appropriate adjustments in the new revision of the manuscript.

Reviewer 6

REVIEWERS' COMMENTS

Reviewer #1 (Remarks to the Author):

I am happy with the response to both my comments and those to the other referees with one major omission. I do not agree with referee 5. When the one point is cleared up the paper should be published, although I remain in disagreement about the use of the word "small polaron" -- that others have misused it for surfaces is not relevant.

The big problem is that their Bader charges in Figure 5 are horribly wrong. They have the oxygens as positive, and the titanium's as negative which is nonsense. I think they have the signs reversed.

Reviewer #2 (Remarks to the Author):

The revised manuscript has fully addressed the questions and concerns raised during the first round of review. I recommend it to be accepted for publication.

Reviewer #3 (Remarks to the Author):

Reviewer #4 (Remarks to the Author):

The authors have diligently addressed all concerns highlighted in the initial report and have effectively incorporated the necessary modifications. Notably, they have appropriately acknowledged the contributions of previous literature and tempered the assertions of novelty. Considering these commendable revisions, we confidently recommend the revised manuscript for publication in Nature Communications.

Reviewer #5 (Remarks to the Author):

Although I`m still not fully convinced about the significance of the manuscript in the current context (most of the concepts are known); the outstanding high quality images, as well as the thorough theoretical analysis performed on this system has an undoubtable fundamental value. The authors have made changes to improve the reading of the manuscript and addressed the queries of all reviewers. I recommend it for publication.

Reviewer #6 (Remarks to the Author):
